# Unveiling the Threat of Maternal Advanced Glycation End Products to Fetal Muscle: Palmitoleic Acid to the Rescue

**DOI:** 10.3390/nu16121898

**Published:** 2024-06-16

**Authors:** Hitomi Yoshizaki, Ritsuko Kawaharada, Saki Tsutsumi, Haruka Okami, Akiyo Toriumi, Eri Miyata, Akio Nakamura

**Affiliations:** 1Department of Bioregulatory Science (Physiology), Nippon Medical School, Tokyo 113-8602, Japan; hitomi-yoshizaki@nms.ac.jp; 2Department of Health and Nutrition, Takasaki University of Health and Welfare, Takasaki 370-0033, Japan; nasu@takasaki-u.ac.jp; 3Department of Neurophysiology & Neural Repair, Graduate School of Medicine, Gunma University, Maebashi 371-8511, Japan; m2310007@gunma-u.ac.jp; 4Department of General Surgical Science, Graduate School of Medicine, Gunma University, Maebashi 371-8511, Japan; m2220010@gunma-u.ac.jp; 5Department of Public Health, Graduate School of Medicine, Gunma University, Maebashi 371-8511, Japan; m2320028@gunma-u.ac.jp; 6Department of Molecular Nutrition, Faculty of Human Life Sciences, Jissen Women’s University, Hino 191-8510, Japan; 2115016m@jissen.ac.jp

**Keywords:** hyperglycemic intrauterine environment, advanced glycation end products, skeletal muscle, reactive oxygen species, palmitoleic acid

## Abstract

Advanced glycation end products (AGEs) accumulate in the plasma of pregnant women with hyperglycemia, potentially inducing oxidative stress and fetal developmental abnormalities. Although intrauterine hyperglycemia has been implicated in excessive fetal growth, the effects of maternal AGEs on fetal development remain unclear. We evaluated the differentiation regulators and cellular signaling in the skeletal muscles of infants born to control mothers (ICM), diabetic mothers (IDM), and diabetic mothers supplemented with either cis-palmitoleic acid (CPA) or trans-palmitoleic acid (TPA). Cell viability, reactive oxygen species levels, and myotube formation were assessed in AGE-exposed C2C12 cells to explore potential mitigation by CPA and TPA. Elevated receptors for AGE expression and decreased Akt and AMPK phosphorylation were evident in rat skeletal muscles in IDM. Maternal palmitoleic acid supplementation alleviated insulin resistance by downregulating RAGE expression and enhancing Akt phosphorylation. The exposure of the C2C12 cells to AGEs reduced cell viability and myotube formation and elevated reactive oxygen species levels, which were attenuated by CPA or TPA supplementation. This suggests that maternal hyperglycemia and plasma AGEs may contribute to skeletal muscle disorders in offspring, which are mitigated by palmitoleic acid supplementation. Hence, the maternal intake of palmitoleic acid during pregnancy may have implications for fetal health.

## 1. Introduction

Clinical studies have long underscored the contribution of gene–environment interactions during fetal and early postnatal periods to non-infectious diseases, with a particular emphasis on the role of the fetal environment, as described in the “Developmental Origins of Health and Disease theory” proposed by Gluckman and Hanson [1]. Although this theory traditionally focuses on the adverse effects of a low-nutritional in utero environment due to maternal malnutrition, recent trends have seen an exponential rise in global diabetes cases to 500 million. The vast number of individuals worldwide consuming excessive amounts of inexpensive sugars has generated a troubling trajectory [2]. Additionally, late marriage and childbearing age have contributed to the prevalence of abnormal glucose metabolism in pregnant women, as well as subsequent gestational diabetes and complications in pregnancies [3]. Maternal hyperglycemia exposes the fetus to a hyperglycemic intrauterine environment via the placenta and contributes to congenital malformations, gigantism, cardiomegaly, metabolic disorders, and several other diseases [4,5,6,7].

Maternal hyperglycemia elevates the levels of advanced glycation end products (AGEs), which are excessively glycated blood proteins [8]. AGEs are generated from various metabolic intermediates, degradation products, and Maillard reaction intermediates apart from glucose [9,10]. Among these, glyceraldehyde-derived AGEs (Glycer-AGEs) are considered to be “toxic AGEs” because of their high cytotoxicity [11]. Glycer-AGEs have been implicated in cell damage and death, and play significant roles in the development and progression of diseases such as cardiovascular disease, nonalcoholic fatty liver disease, Alzheimer’s disease, and cancer [12,13,14,15]. In this study, we focus on two AGE types: glucose-derived AGEs (Glu-AGEs), which are commonly present in patients with diabetes, and Glycer-AGEs, which are intermediates of the polyol metabolic pathway.

AGEs activate receptor for advanced glycation end products (RAGE) signaling, leading to an increase in inflammatory cytokines and reactive oxygen species (ROS) [16]. Previous reports based on animal and cellular models have demonstrated the accumulation of Glu-AGEs in the heart and cerebrum of offspring due to hyperglycemic intrauterine environments, which affects newborns via inflammatory signaling [17,18]. Moreover, excessive AGEs produced by the mother transverse the placenta to reach the fetus and bind to the fetal cell RAGE, potentially impacting fetal development. However, the effects of maternal AGEs on fetal organ development remain poorly understood.

The skeletal muscle is responsible for approximately 80% of blood glucose uptake and plays a crucial role in maintaining normal blood glucose levels [19]. The differentiation process, in particular during embryonic stages and in myosatellite cells, is regulated by a group of Pax genes and myogenic transcription factors that suppress myoblast differentiation [20,21]. The nutrient metabolism in skeletal muscles is regulated by insulin signaling, which promotes glucose uptake and protein synthesis through GLUT4 translocation to the plasma membrane via Akt phosphorylation [22]. An insulin-independent glucose uptake pathway involving AMPK phosphorylation during muscle contraction has also been identified [23,24].

Hyperglycemic intrauterine environments may promote excessive fetal development by upregulating the expression of myogenic transcription factors and Akt phosphorylation in rat skeletal muscle cells, leading to increased differentiation and myotube formation [25]. However, the effects of AGEs supplied to the fetus via the placenta in a hyperglycemic maternal environment remain unclear.

Palmitoleic acid, a functional lipid, can inhibit excessive fetal growth in hyperglycemic intrauterine environments. This molecule, a monounsaturated fatty acid predominantly found in adipose tissue and the liver, has two geometric isomers: the cis-type palmitoleic acid (C16:1 n-7, CPA), which is synthesized in the human body, and the trans-type palmitoleic acid (C16:1 n-7, TPA), which is not endogenously synthesized [26]. CPA, which is derived from the saturated fatty acid palmitate, may positively influence glucose and lipid metabolism [27]. TPA, obtained from dairy products, exerts anti-inflammatory effects in the skeletal muscle, enhances glucose uptake, and improves insulin sensitivity, which suggests a potential role of TPA in reducing the risk of type 2 diabetes. However, regular intake of TPA is essential to maintain plasma TPA concentrations [28,29,30].

To date, the effects of intrauterine hyperglycemia and palmitoleic acid intake on fetal skeletal muscle differentiation in pregnant women remain unexplored. Further, the effects of the potential accumulation of maternally supplied AGEs in fetal skeletal muscle via the placenta are poorly understood. Therefore, the aim of this study is to investigate the effects of intrauterine hyperglycemia on fetal skeletal muscle differentiation in myoblast cells and a rat model of maternal hypoglycemia by examining the effect of AGEs on skeletal muscle differentiation. Additionally, we explore the potential ameliorative effect of palmitoleic acid on the adverse effects exerted by hyperglycemic intrauterine environments on fetal health and assess the potential differences in the beneficial effects of its isomers, CPA and TPA.

## 2. Materials and Methods

### 2.1. Animals

The animal experiments were approved by the Animal Experimentation Committee of Jissen Women’s University (Approval No. 2019-02 and 2021-03). Pregnant Wistar rats (one day of gestation, *n* = 16) were purchased from CLEA Japan (Tokyo, Japan). All rats were housed in a room maintained at 25 °C with 60% humidity and a 12 h light/dark cycle. The rats were fed a normal diet (CE-2, CLEA Japan, Tokyo, Japan) and had free access to water. The rats were randomly divided into four groups: control mothers (CM, *n* = 4), diabetic mothers (DM, *n* = 4), diabetic mothers fed CPA (DM+CPA, *n* = 4), and diabetic mothers fed TPA (DM+TPA, *n* = 4). Diabetes was induced by injecting the rats with 60 mg/kg streptozotocin (STZ, Wako, Osaka, Japan) in 0.05 M citrate buffer (pH 4.5) on day three of gestation [17,18]. Two days after STZ administration, the DM+CPA and DM+TPA groups were fed 150 mg/kg CPA (cis-9-hexadecenoic acid, Tokyo Chemical Industry, Tokyo, Japan) and 150 mg/kg TPA (9(E)-hexadecenoic acid, Larodan AB, Stockholm, Sweden), respectively, using a gastric tube for 16 days. The quadriceps muscles of six infants from each of the CM (ICM, *n* = 6), DM (IDM, *n* = 6), DM+CPA (IDM+CPA, *n* = 6), and DM+TPA (IDM+TPA, *n* = 6) groups were then isolated, where infants were randomly selected from each group.

### 2.2. Cell Culture

C2C12 cells (RIKEN BioResource Research Center Cell Bank, Tsukuba, Japan) were incubated in DMEM (Thermo Fisher Scientific, Waltham, MA, USA) supplemented with 10% FBS (Biowest, Riverside, MO, USA) in a humidified atmosphere with 5% CO_2_ at 37 °C. Upon reaching confluence, 2.5 × 10^3^ cells/cm^2^ and 5.0 × 10^3^ cells/cm^2^ cells were seeded on 96-well microplates and µ-Slide 18 Wells (ibidi, GmbH, Gräfelfing, Germany), respectively.

### 2.3. Preparation of AGEs

Glu-AGEs and Glycer-AGEs were synthesized as previously described [31]. The Glu-AGEs were synthesized by incubating bovine serum albumin (BSA, Sigma-Aldrich, St. Louis, MO, USA) with 0.5 M D-glucose (Wako, Osaka, Japan) and 5 mM diethylenetriaminepentaacetic acid (DTPA, Tokyo Chemical Industry, Tokyo, Japan) in 0.2 M phosphate buffer (pH 7.4) at 37 °C for eight weeks.

To produce the Glycer-AGEs, BSA was incubated with 0.1 M D-glyceraldehyde (Sigma-Aldrich, St. Louis, MO, USA) and 5 mM DTPA in 0.2 M phosphate buffer (pH 7.4) at 37 °C for one week. The reaction products were centrifuged at 10,000× *g* for 30 min in an Amicon Ultra 100 K centrifugal ultrafiltration tube (Merck Millipore, Burlington, MA, USA). Subsequently, low-molecular-weight reactants and residual glucose were eliminated by dialysis in phosphate-buffered saline (PBS) using a Spectra/Por 3 dialysis membrane (Repligen Corporation, Waltham, MA, USA) at pH 7.4.

### 2.4. Preparation of BSA-Conjugated Palmitoleic Acid

CPA (Tokyo Chemical Industry Tokyo, Japan) and TPA (Cayman Chemical Company, Ann Arbor, MI, USA) were conjugated with 10% fatty-acid-free BSA at 4 °C for 2 h. The final concentration of BSA-conjugated CPA and TPA was 4 mM; all mixtures were prepared at the time of use.

### 2.5. Cell Viability Assay

The C2C12 cells were seeded on 96-well microplates and incubated in either DMEM containing 5 mM glucose (low-glucose medium; LG) and 0 (control), 100, 200, 300, or 400 µg/mL of Glu-AGEs or Glycer-AGEs, or DMEM containing 25 mM glucose (high-glucose medium; HG) and 0, 100, 200, or 300 µg/mL of Glu-AGEs or Glycer-AGEs for 72 h. The concentrations of AGEs were chosen based on a previous study [32,33,34]. The AGE cell viability was measured using a Cell Counting Kit-8. Absorbance was measured at 450 nm using a ThermoMax Microplate Reader (Molecular Devices, Ramsey, MN, USA). Cell viability was calculated based on the viability of the cells incubated with or without Glu-AGEs or Glycer-AGEs.

### 2.6. Measurement of Intracellular ROS Generation

Intracellular ROS generation was measured using CellROX^®^ Green Reagent (Thermo Fisher Scientific, Waltham, MA, USA) and 4′,6-diamidino-2-phenylindole (DAPI) (Ibidi Mounting Medium with DAPI, Nippon Genetics, Tokyo, Japan).

C2C12 cells seeded on µ-Slide 18 Wells were incubated with either LG (control), HG, or HG plus 200 µg/mL of Glu-AGEs or Glycer-AGEs for 72 h. To examine the inhibitory effect of palmitoleic acid on intracellular ROS generation, C2C12 cells seeded on µ-Slide 18 Wells were incubated with HG and Glu-AGEs, along with either 200 µM CPA (Glu-AGEs+CPA) or 200 µM TPA (Glu-AGEs+TPA), or Glycer-AGEs and either 200 µM CPA (Glycer-AGEs+CPA) or 200 µM TPA (Glycer-AGEs+TPA) for 72 h. The cells were subsequently incubated with CellROX^®^ Green Reagent (Thermo Fisher Scientific, Waltham, MA, USA) for 30 min and fixed with 3.7% formaldehyde for 15 min. Post washing with PBS, the cells were mounted with DAPI and observed under a BZ-X810 fluorescence microscope (Keyence, Osaka, Japan). Intracellular ROS generation was quantified by calculating the relative ratio of nuclear signals to ROS signals.

### 2.7. Immunofluorescence Staining

C2C12 cells seeded on µ-Slide 18 Wells were incubated with either LG (control), HG, or HG and either Glu-AGEs, Glu-AGEs+CPA, Glu-AGEs+TPA, Glycer-AGEs, Glycer-AGEs+CPA, or Glycer-AGEs+TPA for 72 h, followed by a change to DMEM supplemented with 2% HS and 100 ng/mL of insulin. Differentiated cells were washed with PBS and fixed with 3.7% formaldehyde at 25 °C for 15 min. The cells were then washed with PBS, permeabilized with 0.1% Triton X-100 in 3.7% formaldehyde for 15 min, washed again, and blocked with PBS containing 5% BSA for 1 h. The cells were incubated with an anti-myosin IIb antibody (1:200; #3404) for 1 h, followed by incubation with Alexa Fluor 488 goat anti-rabbit IgG (1:1000) for 1 h at 25 °C. After washing with PBS, the cells were mounted with DAPI and observed under a BZ-X810 fluorescence microscope. The myosin area was quantified by calculating the relative ratio of the nuclear and myosin signals.

### 2.8. RNA Extraction and RT-qPCR

The total RNA was extracted from the quadriceps muscles using the NucleoSpin^®^ TriPrep Total DNA, RNA, and Protein Isolation kit (MACHEREY-NAGEL GmbH & Co., KG, Düren, Germany). cDNA was obtained by reverse transcription of the total RNA using ReverTra Ace^®®^ qPCR RT Master Mix with gDNA Remover (Toyobo, Osaka, Japan). RT-q PCR was performed using a StepOne Plus Real-Time System (Applied Biosystems, Foster City, CA, USA) and KOD SYBR qPCR Mix (Toyobo, Osaka, Japan). The PCR cycle conditions included 98 °C for 2 min followed by 40 cycles of denaturation at 98 °C for 10 s, annealing at 60 °C for 10 s, and extension at 68 °C for 30 s. The primer sequences used for the RT-qPCR are listed in Table 1. The mRNA levels of *Pax3*, *MyoD*, *Myog*, and *Myhc2a* were normalized to the expression of the housekeeping gene, *GAPDH*. The relative levels of mRNA were calculated using the comparative cycle threshold (Ct) (2−ΔΔCt) method.

### 2.9. Western Blotting Analysis

The quadriceps muscles were homogenized and centrifuged using the NucleoSpin^®^ TriPrep Total DNA, RNA, and Protein Isolation kit (MACHEREY-NAGEL GmbH & Co., KG, Düren, Germany) for sample preparation for Western blotting. The protein precipitate was solubilized in Laemmli sample buffer. The protein concentration was determined using a Protein Quantification Assay (MACHEREY-NAGEL GmbH & Co., KG, Düren, Germany). Equal amounts of total protein were separated using sodium dodecyl sulfate–polyacrylamide gel electrophoresis and transferred to polyvinylidene difluoride membranes (Merck Millipore, Burlington, MA, USA). The membranes were blocked with either 5% non-fat dry milk for RAGE, AGEs, and GAPDH or 5% BSA for Akt and AMPK. The membranes were subsequently incubated with primary antibodies against phospho-Akt (Ser473, #9271), Akt (#9272), phospho-AMPKα (#2535), AMPKα (#5831), GLUT4 (#2213), and GAPDH (#5174) purchased from Cell Signaling Technology (Boston, MA, USA), AGEs (KH001) purchased from Trans Genic Inc. (Fukuoka, Japan), and RAGE (ab3611) purchased from Abcam (Cambridge, MA, USA). The membranes were washed with Tris-buffered saline containing 0.1% Tween 20 and incubated with horseradish peroxidase-linked anti-rabbit IgG [#7074] and horseradish peroxidase-linked anti-mouse IgG [#7076] purchased from Cell Signaling Technology. Protein bands were detected by an Odyssey^®^ Fc Imaging System (LI-COR Biosciences, Lincoln, NE, USA).

### 2.10. Statistical Analysis

All data are expressed as the mean ± standard error of the mean (SEM). Statistical significance was analyzed using Student’s *t*-test. Differences were considered to be statistically significant at *p* < 0.05.

## 3. Results

### 3.1. Effect of Palmitoleic Acid Intake on Gene Expression in Fetal Skeletal Muscle in Hyperglycemic Intrauterine Environments

The effects of maternal hyperglycemia on skeletal muscle differentiation were assessed by examining the mRNA expression associated with this process in fetal skeletal muscle samples from the ICM, IDM, IDM+CPA, and IDM+TPA groups. Although the levels of *Pax3* mRNA, which inhibits differentiation and maintains the undifferentiated state, were 1.5 times higher in the IDM group than in the ICM group, they were 0.5 times lower in the IDM+CPA group than in the IDM group. Notably, no significant differences in *Pax3* expression were observed between the IDM and IDM+TPA groups (Figure 1A). The mRNA expression of *MyoD*, a regulator of the initial stages of muscle differentiation, was 0.7 times lower in the IDM group than in the ICM group, and 1.9 times higher in the IDM+CPA group than in the IDM group. No significant differences in *MyoD* expression were observed between the IDM and IDM+TPA groups (Figure 1B). The mRNA expression of *Myog*, which regulates late muscle differentiation, was 0.7 times lower in the IDM group than in the ICM group, and 2.4 and 1.6 times higher in the IDM+CPA and IDM+TPA groups than in the IDM group, respectively (Figure 1C). The mRNA expression of *Myhc2a*, a marker of the final stage of differentiation, was 0.3 times higher in the IDM group than in the ICM group, and 7.3 and 4.4 times higher in the IDM+CPA and IDM+TPA groups than in the IDM group, respectively (Figure 1D).

### 3.2. Effects of Maternal Hyperglycemia and Intake of Palmitoleic Acid on Protein Expression and Phosphorylation in Fetal Skeletal Muscle

The protein expression and phosphorylation in fetal skeletal muscles of the ICM, IDM, IDM+CPA, and IDM+TPA groups were analyzed to assess the impact of maternal hyperglycemia on insulin and oxidative stress signaling. RAGE expression was 2.0 times higher in the IDM group than in the ICM group, but 0.6 and 0.5 times lower in the IDM+CPA and IDM+TPA groups than in the IDM group, respectively (Figure 2A). Akt phosphorylation was 0.5 times lower in the IDM group than in the ICM group, but 1.4 and 1.6 times higher in the IDM+CPA and IDM+TPA groups than in the IDM group, respectively (Figure 2B). No significant differences in GLUT4 expression were observed between the groups (Figure 2C). AMPKα phosphorylation was 0.4 times lower in the IDM group than in the ICM group, but 1.3 times higher in the IDM+TPA group than in the IDM group. Notably, no significant differences in AMPKα phosphorylation were observed between the IDM and IDM+CPA groups (Figure 2D). Moreover, the extent of AGE formation was 0.4 times lower in the IDM group than in the ICM group, but 1.5 times higher in the IDM+TPA group than in the IDM group. No significant differences in AGE formation were observed between the IDM and IDM+CPA groups (Figure 2E).

### 3.3. Viability of C2C12 Cells Treated with AGEs

Cell viability was evaluated as a measure of AGE cytotoxicity in C2C12 cells. The C2C12 cell viability decreased in a dose-dependent manner after treatment with AGEs for 72 h (Figure 3A–D). Compared with the control, the C2C12 cells treated with LG and Glu-AGEs exhibited a 14% decrease in viability at a concentration of 300 µg/mL (Figure 3A). The viability of the C2C12 cells treated with LG and Glycer-AGEs exhibited a greater decrease of 24% at a concentration of 300 µg/mL than that exhibited by the control (Figure 3B). Compared with the control, the C2C12 cells treated with HG and either Glu-AGEs or Glycer-AGEs demonstrated decreases in cell viability of 21% and 33%, respectively, at a concentration of 200 µg/mL (Figure 3C,D).

### 3.4. Intracellular ROS Generation Level in AGE-Treated C2C12 Cells

Intracellular ROS levels were measured 72 h after treatment with AGEs and palmitoleic acid to examine AGE-induced intracellular ROS generation, as well as the inhibition of the same by palmitoleic acid in the C2C12 cells. Intracellular ROS generation was 2.4 and 2.2 times higher in the C2C12 cells treated with Glu-AGEs and Glycer-AGEs, respectively, than in the control. However, the addition of CPA or TPA to C2C12 cells treated with Glu-AGEs and Glycer-AGEs resulted in a decrease in intracellular ROS generation by 0.6-fold and 0.5-fold, respectively (Figure 4A–C).

### 3.5. Myotube Formation in AGE-Treated C2C12 Cells

C2C12 cells were treated with HG and AGEs alone or in combination with palmitoleic acid to examine the effects of AGEs on myotube formation through myosin signals. Despite observing no significant differences, myotube formation in the C2C12 cells treated with HG and Glu-AGEs increased from that in the control cells (Figure 5A,B). However, compared with the control, treatment with Glycer-AGEs decreased myotube formation by 0.3-fold. Further, the addition of palmitoleic acid to Glycer-AGEs did not promote myotube formation (Figure 5A,C).

## 4. Discussion

In this study, we investigated the effects of maternal hyperglycemia on fetal skeletal muscle using a rat model of abnormal glucose metabolism. Additionally, we explored the potential benefits of administering palmitoleic acid, an n-7 unsaturated fatty acid, during gestation. A schematic summarizing the results of this study is shown below (Figure 6). Compared with the ICM group, the expressions of *Pax3* and *MyoD* were significantly increased and significantly decreased, respectively, in the skeletal muscles of the IDM group exposed to maternal hyperglycemic conditions. *Pax3* inhibits differentiation and maintains an undifferentiated state, whereas *MyoD* serves as a myogenic marker that facilitates early differentiation [20,35]. Moreover, *Pax3* expression gradually declines as skeletal muscle differentiation progresses, whereas *MyoD* expression increases [36]. Our findings suggest that the upregulation of *Pax3* expression in the skeletal muscles of the IDM group hindered *MyoD* expression, thereby suppressing early skeletal muscle differentiation. Although we observed no changes in gene expression in the IDM group supplemented with TPA, a significant reduction in *Pax3* expression and increase in *MyoD* expression were observed in the IDM group supplemented with CPA. This implies that, although TPA intake does not alleviate the hyperglycemic-environment-induced suppression of early skeletal muscle differentiation, CPA administration may mitigate this effect. The expressions of *Myog* and *Myhc2a*, which are associated with late skeletal muscle differentiation, were notably decreased in the skeletal muscle of the IDM-alone group, but significantly increased in the IDM groups supplemented with CPA and TPA. These results indicate that both CPA and TPA may mitigate the hyperglycemic-environment-induced suppression of late skeletal muscle differentiation. Thus, maternal CPA intake during gestation may alleviate the hyperglycemic-environment-mediated inhibition of both early and late skeletal muscle differentiation. In contrast, TPA intake may mitigate the hyperglycemic-environment-induced inhibition of skeletal muscle differentiation in the late phase rather than in the early phase. Thus, the intake of cis- and trans-palmitoleic acid during gestation may exert distinct effects.

Signaling and protein expression analyses revealed that the RAGE expression and Akt phosphorylation levels in the IDM group were significantly elevated and reduced, respectively, compared to those in the ICM group. These findings suggest that maternal hyperglycemic environments augment plasma AGE–RAGE signaling that leads to insulin resistance via ROS production and inflammatory signaling. RAGE expression can be further increased via a positive feedback route through elevated ROS and NF-κB activation associated with inflammatory signaling, which indicates the initiation of a potentially detrimental cycle [37,38,39]. Additionally, AMPK phosphorylation was diminished in the IDM group, akin to Akt, indicating the inhibition of both pathways, which would facilitate glucose uptake into skeletal muscle cells. AMPK activation typically occurs during muscle contraction, when ATP generated from the glycolytic system is converted into AMP. The diminished phosphorylation of AMPK in the IDM group may, thus, be attributed to inhibited glucose uptake into cells caused by insulin resistance [24]. This correlates with our finding of significantly reduced intracellular AGE production in the IDM group in a hyperglycemic environment, which may be a consequence of impaired glucose uptake into skeletal muscle cells following insulin resistance. Notably, compared with the IDM-alone group, the RAGE expression was diminished and the Akt phosphorylation levels were significantly elevated in the IDM groups supplemented with CPA and TPA. This implies that palmitoleic acid intake may attenuate AGE–RAGE signaling and enhance insulin sensitivity. Two plausible explanations exist for this observation. First, cellular models only simulate a transient hyperglycemic environment because of the challenges of long-term cultures; therefore, the upregulation of gene expression involved in skeletal muscle differentiation and Akt phosphorylation in these models may be attributable to nutritional metabolism signaling subsequent to this effect. Second, maternal blood contains AGEs produced because of abnormalities in the maternal glucose metabolism, which can adversely affect the fetus upon transversing the placenta. Hence, the effects of maternal hyperglycemia during pregnancy on the next generation can be better elucidated in the current animal model than in a cellular model. Our findings suggest that the inhibition of fetal skeletal muscle differentiation and the adverse effects on nutritional signaling by maternal hyperglycemia were mitigated by the ingestion of palmitoleic acid.

The maternal plasma concentrations of AGEs in pregnant women with an abnormal glucose metabolism are elevated compared with those in women with normal gestational glucose levels [8]. Hence, maternal hyperglycemia and elevated AGEs may contribute in tandem to known adverse effects. Given the challenge of isolating the effects of AGEs from those of maternal hyperglycemia in animal models, experiments were also conducted using cellular models. Rat L6 and mouse C2C12 cells are frequently used for skeletal myoblast experiments; however, the latter exhibit more stable proliferation, differentiation, and myotube formation in both serum-containing and serum-free culture media. Moreover, C2C12 cells express abundant genes and proteins that are pertinent to muscle development and contraction [40,41].

In this study, we utilized a cellular model that simulated hyperglycemia and AGE exposure using C2C12 cells to investigate their effects on myoblasts. Regardless of LG or HG, exposure to both Glu-AGEs and Glycer-AGEs resulted in a concentration-dependent reduction in cell viability, with higher AGE concentrations proving to be cytotoxic to myoblasts. As well as differing in their sugar sources, Glycer-AGEs are distinct from Glu-AGEs on account of being more potent inducers of cellular damage, and are implicated in the pathogenesis of various diseases [42,43,44]. We observed no cytotoxicity of LG in combination with Glu-AGEs up to a concentration of 200 µg/mL, and the same was evident with LG and 100 µg/mL of Glycer-AGEs. Further, in HG, the cytotoxicity of both AGEs was evident at concentrations above 200 µg/mL. Thus, it is plausible that, in the presence of HG, Glycer-AGE exposure initially boosts glucose metabolism and cell viability, which decreases after a gradual induction of insulin resistance.

Although AGE cytotoxicity studies have been conducted in various cell types, few studies have explored the cytotoxicity of distinct AGE types in specific cells. Here, we found that Glycer-AGEs are more cytotoxic to C2C12 cells than Glu-AGEs. A previous study reported a significant reduction in cell viability (approximately 70%) following exposure to 800 µg/mL of Glycer-AGEs for 24 h in H9C2 cells compared with that in control cells [33]. Additionally, in the human fetal osteoblastic cell line hFOB1.19, exposure to 150 µg/mL of Glu-AGEs for 24 h transiently increased osteoblast cell viability by approximately 150% from that in the controls, suggesting enhanced bone remodeling due to excessive bone formation and subsequent imbalance. Conversely, exposure to 200 µg/mL of Glu-AGEs for 24 h significantly decreased cell viability by approximately 20%, indicating osteoblast apoptosis [34]. These findings indicate that the toxic concentrations and durations of exposure of AGEs vary by the type of cell line, and that AGE exposure may not solely reduce cell viability and function. Maternally supplied AGEs induce inflammatory signals and cause cell death via AGE–RAGE signaling [45]. Similarly, we observed a significant reduction in cell viability under HG conditions with AGEs, which highlights the detrimental effects of maternally derived AGEs on myoblasts.

Excessive intracellular ROS production may deactivate the antioxidant system and exacerbate glycation stress [46]. In this study, we observed increased intracellular ROS levels following exposure to HG and AGEs. This suggests that hyperglycemia and plasma AGEs may induce inflammatory signals via increased intracellular ROS levels, which lead to skeletal muscle tissue damage. CPA or TPA supplementation in the presence of Glu-AGEs or Glycer-AGEs reduced intracellular ROS generation across all groups, which indicates that both cis- and trans-palmitoleic acid exert an inhibitory effect on intracellular ROS generation. Traditionally, ROS has been viewed solely as being detrimental to skeletal muscle tissue, as it contributes to oxidative stress, pathogenesis, and aging. However, recent studies have suggested more nuanced roles for ROS in skeletal muscle tissue. For example, at low concentrations, ROSs serve as beneficial signaling molecules that regulate physiological processes via the activation of crucial signaling pathways, such as those involving PGC-1α and MAPK [47,48,49,50]. Moreover, intracellular ROSs are produced in the skeletal muscles during exercise-induced loading [51,52,53]. Consequently, an increase in intracellular ROS may not invariably have adverse effects on the skeletal muscle. Further investigations are warranted to elucidate the physiological functions of intracellular ROSs in skeletal muscles, with a specific focus on their impact on myoblast signaling. Further, the mechanisms underlying the suppression of intracellular ROS generation by palmitoleic acid require further elucidation.

Myotube formation is promoted under hyperglycemic conditions in L6 cells [25]. In this study, we observed the inhibition of myotube formation in C2C12 cells exposed to Glycer-AGEs, unlike in the controls, with no significant improvement observed upon the addition of palmitoleic acid. This suggests that myoblast differentiation and myotube formation may be impeded in maternal hyperglycemic environments with accumulated Glycer-AGEs. Moreover, myotube formation in L6 cells under hyperglycemic conditions is impeded by palmitoleic acid, suggesting its putative role in muscle hyperplasia reduction. The inhibition of myotube formation highlights the potentially negative impact of AGEs produced within a hyperglycemic environment. Further, although the addition of TPA in the presence of Glycer-AGEs significantly increased myotube formation, it was not restored to control levels. As Glycer-AGEs exhibit greater toxicity than Glu-AGEs during myoblast growth, the inability of palmitoleic acid to fully restore myotube formation to control levels may be attributed to potential cell death in the early stages of myotube formation. In this study, the gene expression results from both animal and cellular models were in agreement; that is, the expression of genes implicated in muscle differentiation was diminished in the IDM group, and Glycer-AGEs significantly impeded myotube formation in cells. Thus, similar to previous reports, our study showed that the toxicity of Glycer-AGEs surpasses that of Glu-AGEs and may negatively impact myogenesis.

## 5. Conclusions

Our findings reveal that fetal skeletal muscle differentiation is impaired and insulin resistance is induced in an intrauterine hyperglycemic environment. These effects are associated with the production of plasma AGEs, which may lead to skeletal muscle tissue damage. As the addition of palmitoleic acid mitigated these effects, we suggest that the maternal intake of palmitoleic acid during gestation may benefit fetal health.

This study is limited by the fact that the results were obtained in rat and cellular models, so may not be immediately applicable to humans. Additionally, the detailed molecular mechanisms underlying the inhibitory effects of palmitoleic acid on inflammatory signaling remain unclear. Further, the epigenetic modifications inscribed in the genomic DNA following maternal hyperglycemia and AGE production during the fetal period, as well as the subsequent impact on skeletal muscle function in the offspring, are not well understood. Future studies should investigate the detailed molecular mechanisms underlying the suppression of inflammatory signals by palmitoleic acid and the effects of maternal hyperglycemia and AGE production on fetal skeletal muscle differentiation via epigenomic analyses.

## Figures and Tables

**Figure 1 nutrients-16-01898-f001:**
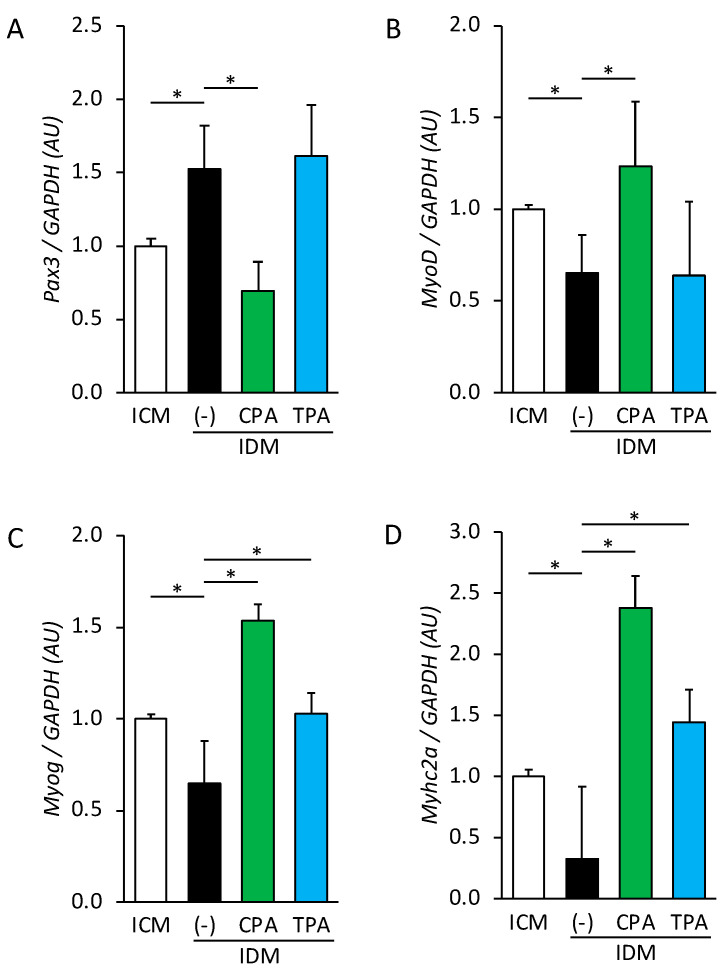
mRNA expression levels of *Pax3* (**A**), *MyoD* (**B**), *Myog* (**C**), and *Myhc2a* (**D**) in fetal skeletal muscles from ICM, IDM, IDM+CPA, and IDM+TPA groups. mRNA levels were normalized to those of *GAPDH*. Data are represented as the mean ± SEM, *n* = 6. * *p* < 0.05.

**Figure 2 nutrients-16-01898-f002:**
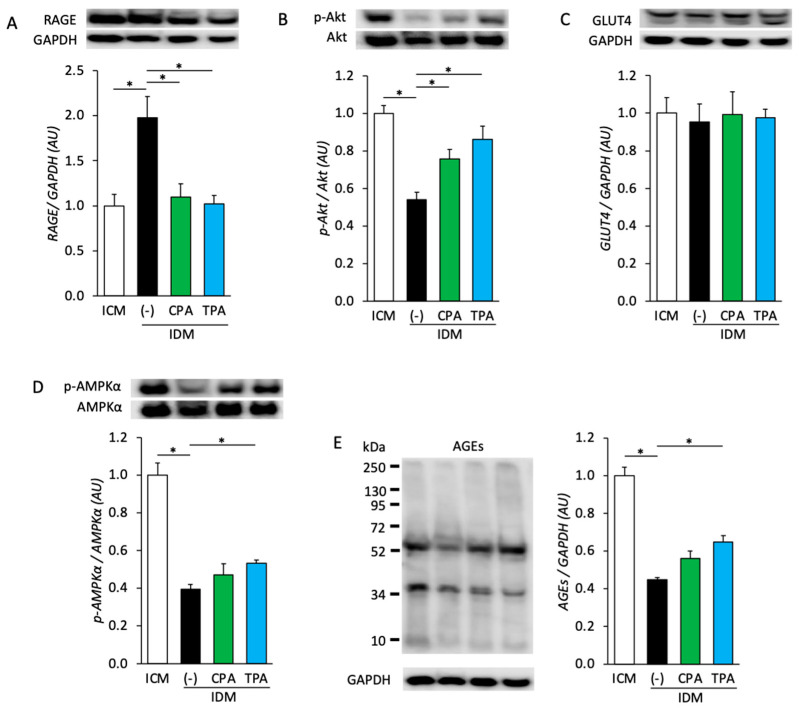
Protein expression and phosphorylation in fetal skeletal muscle from ICM, IDM, IDM+CPA, and IDM+TPA groups. (**A**) RAGE expression, (**B**) Akt phosphorylation, (**C**) GLUT4 expression, (**D**) AMPKα phosphorylation, and (**E**) AGE levels were assessed. Data are represented as the mean ± SEM, *n* = 6. * *p* < 0.05.

**Figure 3 nutrients-16-01898-f003:**
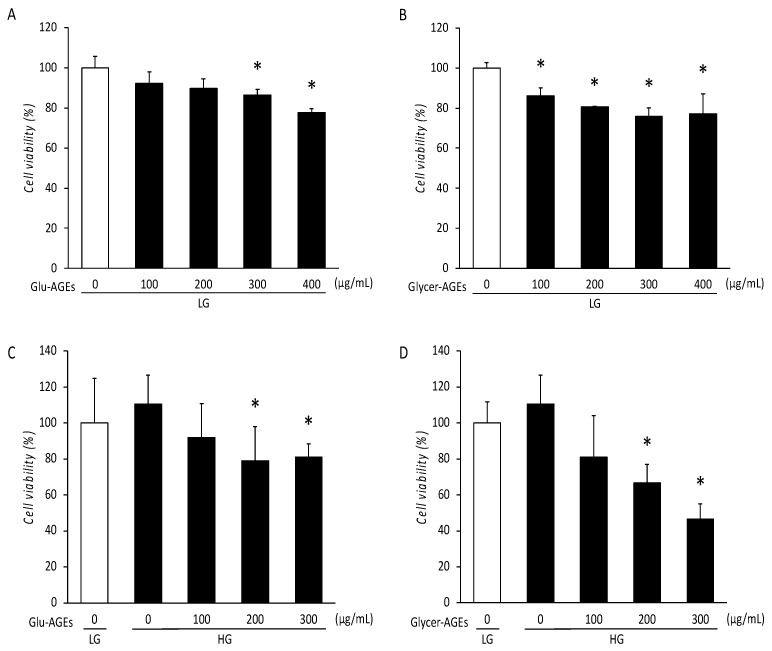
Viability of C2C12 cells treated with AGEs for 72 h. Cell viability was assessed using the WST-8 assay. (**A**) C2C12 cells were incubated with LG (5 mM glucose) and 0 (control), 100, 200, 300, or 400 µg/mL Glu-AGEs for 72 h. (**B**) C2C12 cells were incubated with LG and 0 (control), 100, 200, 300, or 400 µg/mL Glycer-AGEs for 72 h. (**C**) C2C12 cells were incubated with LG without AGEs (control), HG, and 0, 100, 200, or 300 µg/mL Glu-AGEs for 72 h. (**D**) C2C12 cells were incubated with LG without AGEs (control), HG (25 mM glucose), and 0, 100, 200, or 300 µg/mL Glycer-AGEs for 72 h. Data are represented as the mean ± SEM, *n* = 5. * *p* < 0.05 vs. control.

**Figure 4 nutrients-16-01898-f004:**
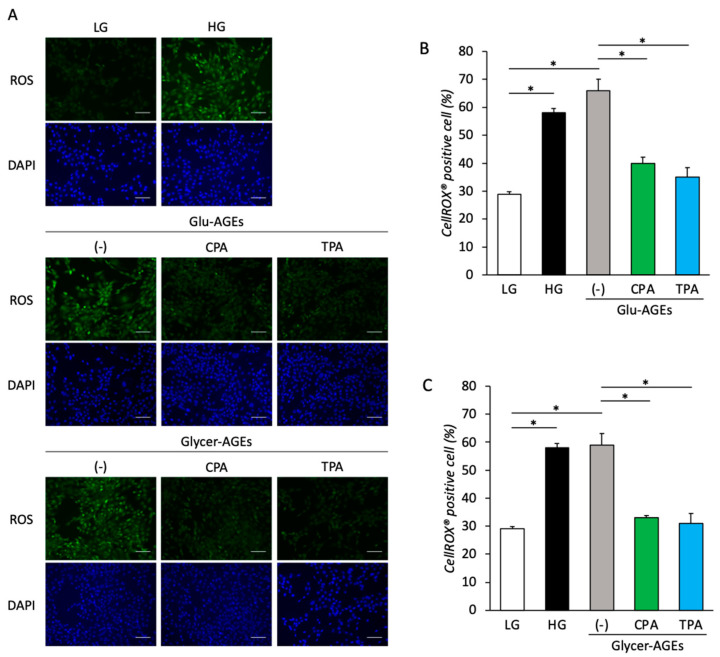
Intracellular ROS generation was assessed by incubating C2C12 cells with AGEs (200 µg/mL) alone or in combination with palmitoleic acid (200 µM) for 72 h. (**A**) Images showing ROS (green) and DAPI (blue) in C2C12 cells treated with LG (5 mM glucose, control), HG (25 mM glucose), Glu-AGEs, Glu-AGEs+CPA, Glu-AGEs+TPA, Glycer-AGEs, Glycer-AGEs+CPA, and Glycer-AGEs+TPA for 72 h. Scale bars are 100 µm. (**B**) Intracellular ROS generation in C2C12 cells treated with Glu-AGEs, Glu-AGEs+CPA, and Glu-AGEs+TPA was quantified by determining the relative ratio of nuclear and ROS signals based on the data in (**A**). (**C**) Intracellular ROS generation in C2C12 cells treated with Glycer-AGEs, Glycer-AGEs+CPA, and Glycer-AGEs+TPA was quantified by determining the relative ratio of nuclear and ROS signals based on the data in (**A**). Data are represented as the mean ± SEM, *n* = 5. * *p* < 0.05.

**Figure 5 nutrients-16-01898-f005:**
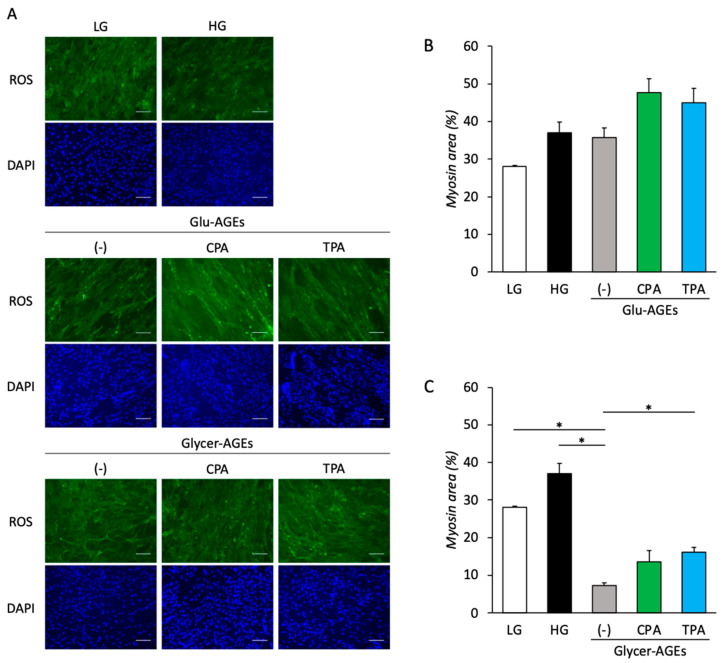
Myotube formation was assessed by detecting myosin signals in C2C12 cells treated with HG and AGEs alone or in combination with palmitoleic acid. (**A**) Images showing myosin (green) and DAPI (blue) in C2C12 cells treated with LG (5 mM glucose, control), HG (25 mM glucose), Glu-AGEs, Glu-AGEs+CPA, Glu-AGEs+TPA, Glycer-AGEs, Glycer-AGEs+CPA, and Glycer-AGEs+TPA. Scale bars are 100 µm. (**B**) Myosin areas in HG, Glu-AGEs, Glu-AGEs+CPA, and Glu-AGEs+TPA were quantified by determining the relative ratio of nuclear and myosin signals based on the data in (**A**). (**C**) Myosin areas in HG, Glycer-AGEs, Glycer-AGEs+CPA, and Glycer-AGEs+TPA were quantified by determining the relative ratio of nuclear and myosin signals based on the data in (**A**). Data are represented as the mean ± SEM, *n* = 5. * *p* < 0.05.

**Figure 6 nutrients-16-01898-f006:**
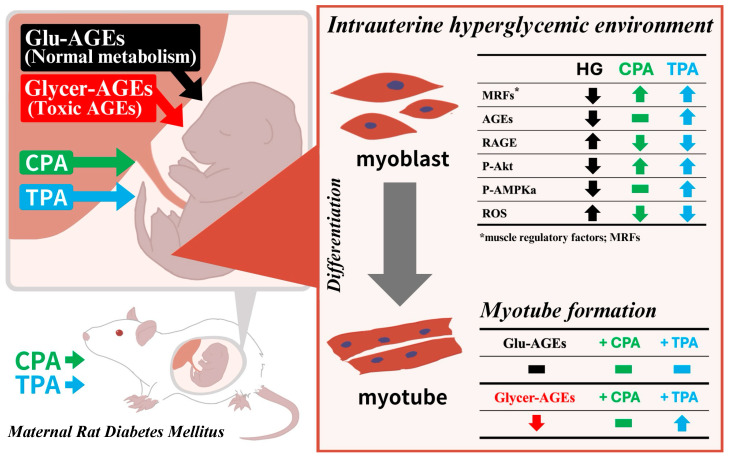
Schematic summarizing the findings of this study. If the mother is diabetic during pregnancy, not only maternal hyperglycemia but also AGEs produced by the mother pass through the placenta, resulting in inhibition of skeletal muscle differentiation and insulin resistance in the fetus. Glycer-AGEs are particularly cytotoxic and inhibit myotube formation. However, these abnormalities may be ameliorated by the intake of palmitoleic acid during pregnancy.

**Table 1 nutrients-16-01898-t001:** Primer sequences used in this study.

Gene	Forward Primer (5′ to 3′)	Reverse Primer (5′ to 3′)
*Pax3*	CAGCCCACGTCTATTCCACA	CACGAAGCTGTCGGTGTAGC
*MyoD*	TGGATCAATCCCACTCTAATAGC	TTCGCTGGTAGGAAAGTGAAG
*Myog*	CTACAGGCCTTGCTCA	TGGGAGTTGCATTCAC
*Myhc2a*	TCCTCAGGCTTCAAGATTTG	TTAAATAGAATCACATGGGGAC
*GAPDH*	CTACCCACGGCAAGTTCAAC	CCAGTAGACTCCACGACATAC

## Data Availability

All data generated or analyzed during this study are included in this published article.

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
