# Peer review of "Unveiling the Threat of Maternal Advanced Glycation End Products to Fetal Muscle: Palmitoleic Acid to the Rescue"

_nutrients, 2024, doi:10.3390/nu16121898_

Round 1

Reviewer 1 Report

Comments and Suggestions for Authors

1. The methods section is detailed, which is excellent for reproducibility. However, some sections could be streamlined. For example, the preparation of AGEs and cell culture conditions could be summarized more concisely. Ensure all critical information is retained but avoid unnecessary details.

2. The discussion is thorough but could be more focused. Highlight the most significant findings and their implications first, then discuss the broader context and potential limitations. This structure will make the discussion more impactful and easier to follow.

3. The conclusion should succinctly summarize the main findings and their significance. Additionally, clearly outline future research directions and potential applications of your findings. This will provide a strong ending to your manuscript and highlight its contribution to the field.

4. Besides studying Glu-AGEs and Glycer-AGEs, test other common AGEs types like methylglyoxal-AGEs (MGO-AGEs) to compare their effects on muscle cells.

5. Conduct more detailed dose-response curves for various concentrations of AGEs to determine their cytotoxicity and impacts at different levels.

6. Test other anti-AGEs substances (such as vitamin C, alpha-lipoic acid) for their protective effects and compare them with palmitoleic acid.

7. Investigate whether anti-inflammatory drugs can reduce inflammation and oxidative stress induced by AGEs, thus protecting muscle cells.

8. Clearly state the hypothesis and specific objectives at the end of the introduction. This will help set the stage for the study and clarify its aims.

9. Provide a concise summary of the current state of research on AGEs, hyperglycemia during pregnancy, and their impacts on fetal development. Highlight key findings and gaps in the literature that your study aims to address.

10. Further elaborate on the Developmental Origins of Health and Disease (DOHaD) hypothesis and its relevance to your study. Discuss how this hypothesis frames your research within a broader context.

11. Clearly describe the specific conditions under which C2C12 cells were cultured, including medium composition, seeding density, and differentiation protocols. 

12. Ensure that all figures and tables are clearly labeled and referenced in the text. Add descriptions in figure legends to make them self-explanatory. Consider adding more visual representations, such as dose-response curves, to enhance data presentation.

13. Offer a more in-depth interpretation of the results. Discuss how your findings align with or differ from previous studies and what this means for the field.

14. Provide more insights into the potential mechanisms by which AGEs and hyperglycemia impact fetal muscle development. Link your findings to specific signaling pathways and molecular processes.

15. Discuss the potential clinical relevance of your findings. Highlight how maternal intake of palmitoleic acid could be a practical intervention to mitigate the adverse effects of hyperglycemia and AGEs on fetal development.

Author Response

We sincerely thank you for taking the time to read our manuscript and provide valuable feedback. Your insightful comments have helped improve the quality of our manuscript. We agree with your suggestions and have diligently revised the manuscript based on your insightful comments.

Reviewer 2 Report

Comments and Suggestions for Authors

The introduction you provided is detailed and covers a wide range of topics related to fetal undernutrition, hyperglycemia, advanced glycation end products (AGEs), and skeletal muscle development. However, there are several flaws and areas that could be improved for better clarity and coherence. Here are some identified issues and suggestions for improvement:

1.     The introduction is filled with complex sentences and technical jargon, which can make it difficult for readers to follow. Simplifying some sentences and defining technical terms could enhance readability.

2.     The introduction jumps between various topics without clear transitions. This can confuse readers about the main focus of the study. A more structured approach with clear subsections or a logical flow of ideas would help.

3.     Some sentences are overly detailed or repetitive. Condensing information to focus on key points will make the introduction more concise.

4.     The introduction introduces multiple concepts (e.g., fetal undernutrition, hyperglycemia, AGEs, skeletal muscle development) without clearly tying them together initially. Establishing a clear connection between these concepts early on is important.

5.     There are many references in close succession. While referencing is important, clustering too many citations can disrupt the flow of reading. Integrating references more seamlessly into the text could help.

The methods described in the provided text outline various experimental procedures involving animal models and cell culture techniques. Here are some potential flaws or areas for improvement in these methods:

1.     While the experiments were approved by an ethical committee, there is always a need for rigorous assessment to ensure the welfare of the animals, minimizing suffering, and considering alternatives.

2.     The method of inducing diabetes using streptozotocin (STZ) can lead to variability in the severity of diabetes among different rats, which might affect the consistency of the results.

3.     It is essential to ensure that control and experimental groups are housed under identical conditions to minimize environmental variables affecting the outcomes.

4.     The use of C2C12 cells, which are a mouse myoblast cell line, might not fully represent the behavior of human muscle cells. It is important to acknowledge this limitation when interpreting the results.

5.     The synthesis of AGEs over an extended period (8 weeks for Glu-AGEs) might lead to batch-to-batch variability, affecting reproducibility.

6.     The removal of low-molecular-weight reactants and residual glucose is crucial, but the efficiency of this process needs to be validated to ensure the purity of the AGEs.

7.     The stability of the BSA-conjugated fatty acids (CPA and TPA) during storage at 4°C needs to be confirmed to ensure consistent experimental conditions.

8.     The chosen concentrations of AGEs (0-400 µg/mL) need to be justified based on preliminary studies to ensure they are within the biologically relevant range.

9.     The method of quantifying ROS generation by calculating the relative ratio of nuclear signals to ROS signals might lack precision. More quantitative approaches, such as flow cytometry, could be considered.

Comments on the Quality of English Language

Minor editing of english language needed

Author Response

We thank you for reading the manuscript and for your valuable comments. We fully agree with your remarks and have revised the manuscript accordingly.   Please see the attachment.

Round 2

Reviewer 1 Report

Comments and Suggestions for Authors

I have no comments